# Metabolic and Functional Improvements in a Patient with Charcot-Marie-Tooth Disease Type 2 after EGCG Administration: A Case Report

**DOI:** 10.3390/medicina57020104

**Published:** 2021-01-24

**Authors:** Antonio Bustos, Pablo Selvi Sabater, María Benlloch, Eraci Drehmer, María Mar López-Rodríguez, Felix Platero, Jose Luis Platero, Jesús Escribá-Alepuz, Jose Enrique de la Rubia Ortí

**Affiliations:** 1Physical Therapy Clinic, Antonio Bustos, 46007 Valencia, Spain; fisioterapiabustos@gmail.com; 2Hospital de Riotinto, 21660 Huelva, Spain; pablo.selvi.sspa@juntadeandalucia.es; 3Department of Nursing, Catholic University of Valencia San Vicente Mártir, 46001 Valencia, Spain; joseluis.platero@mail.ucv.es (J.L.P.); joseenrique.delarubi@ucv.es (J.E.d.l.R.O.); 4Department of Basic Sciences, Catholic University of Valencia San Vicente Mártir, 46900 Torrente, Valencia, Spain; eraci.drehmer@ucv.es; 5Department of Nursing, Physiotherapy and Medicine, University of Almería, 04120 Almería, Spain; mlr295@ual.es; 6Department of Medicine, University of Valencia, 46003 Valencia, Spain; felixplateroarmero@gmail.com; 7Neurophysiology Department, Sagunto University Hospital, 46520 Valencia, Spain; jesusescriba@hotmail.com; 8Institute of Sleep Medicine, 46021 Valencia, Spain

**Keywords:** Charcot-Marie-Tooth disease type 2, epigallocatechin gallate, IL-6, motor activity, paraoxonase 1

## Abstract

*Background and objectives:* The aim of this study was to report a case of a patient with Charcot-Marie-Tooth disease type 2 (CMT2) treated with epigallocatechin gallate (EGCG) for 4 months in order to assess its therapeutic potential in CMT2. *Materials and Methods:* The study included a brother and a sister who have CMT2. The sister received 800 mg of EGCG for 4 months, while her brother received placebo for the same period of time. Both participants were assessed before and after daily administration by means of anthropometry; analysis of inflammatory and oxidation markers of interleukin-6 (IL-6) and paraoxonase 1 (PON1) in the blood sample; and motor tests: 2-min walk test (2MWT), 10-m walk test (10MWT), nine-hole peg test (9HPT) and handgrip strength measurement using a handheld Jamar dynamometer. *Results:* Regarding muscular and motor functions associated with higher inflammation and oxidation, improvements only observed in the woman in all analysed parameters (both biochemical and clinical associated with the metabolism and functionality) after 4 months of treatment with EGCG are noteworthy. Thus, this treatment is proposed as a good candidate to treat the disease.

## 1. Introduction

Charcot-Marie-Tooth disease (CMT) is the most common hereditary neurological disorder. It is characterised by deterioration in the functioning of peripheral nerves, with distal predominance, classified in two types according to the nature of the neuronal alteration: CMT type 1 (CMT1), also known as demyelinating CMT, and CMT type 2 (CMT2) where mainly axon damage occurs [1]. CMT causes progressive bilateral and symmetrical muscular atrophy and weakness of the lower limbs, especially in the muscles in feet and legs, later reaching the upper limbs and affecting forearms and hands with loss of strength and manual dexterity [2].

There is currently no pharmacological cure and the only treatment available is rehabilitative therapy. However, chronic inflammation [3,4] that develops as the disease progresses suggests that treatment with effective anti-inflammatory could improve prognosis. In this sense, epigallocatechin gallate (EGCG), the main polyphenol in green tea, is characterised by its high anti-inflammatory capacity, shown to be especially effective in improving mitochondrial activity by regulating its metabolism, leading to an increase in biogenesis and bioenergy [5].

Regarding the relation of this catechin with body composition, obesity associated with hypertrophy and hyperplasia of the adipose tissue generates inflammation causing oxidative muscular stress, therefore causing more inflammation which deregulates the myocyte metabolism due to mitochondrial dysfunction. In this sense, EGCG decreases oxidative stress and inflammation related to the metabolic dysfunction of skeletal muscle in obesity, as the expression of antioxidant enzymes is increased. As a result, the production of reactive oxygen species (ROS) is reduced and autophagy related to mitochondria is regulated [6].

Paraoxonase 1 (PON1) is an enzyme that inhibits oxidation of LDL, preventing the production of cytokines, inflammatory mediators and cell adhesion molecules. Thus, inflammatory response and deposition of LDL in the blood vessels is reduced [7]. A decrease in PON1 is correlated with the development of some pathologies with high oxidative stress and inflammation [8,9], therefore, it is an effective inflammation marker related to metabolic changes [10,11].

Finally, IL-6 is a cytokine whose high levels indicate inflammation [12]. In this sense, it has very high levels in neuromuscular diseases with a high oxidative stress as in amyotrophic lateral sclerosis (ALS) [13] or muscular dystrophy [14], indicating muscular destruction in these pathologies.

Based on the above, the aim of this study was to report a case of a patient with CMT2 treated with EGCG for four months in order to assess its therapeutic potential in CMT2.

## 2. Materials and Methods

Treatment procedures for the patients were carried out in accordance with the Helsinki Declaration and were approved on 14 December 2017 by the Ethics Committee of the University of Valencia with code number H1512345043343. Both participants gave written consent to publish their case and data in an international medical journal.

Intervention

EGCG was administered to the female patient for 4 months, distributed in two doses of 400 mg (capsules at breakfast and dinner) and following an isocaloric Mediterranean diet (approx. 2300 kcal/day; 55% carbohydrates + 30% fat + 15% proteins). The second patient was administered placebo capsules containing microcrystalline cellulose, matching in size and colour. He was told to follow the same instructions: take two capsules a day (breakfast and dinner) and follow the same Mediterranean diet.

Measurements

The following measurements were taken before and after the 4-month intervention, in the same conditions and at the same time. In the specific case of the test, they were carried out by the same neurologist assigned to each patient before the study.

Body composition: Measurements related to weight, size, skin folds and body perimeters and diameters were taken using the Faulkner method, taking into account the protocol currently established by The International Society for the Advancement of Kinanthropometry (ISAK)SECA [15]. The equipment used for these measurements were: a portable clinical scale, SECA model, with a 150–200 kg capacity and 100 g precision, stadiometer, model SECA 220 (Hamburg, Germany) with 0.1 cm precision, a mechanical skinfold caliper, model Holtain LTD Crymych (Pembrokeshire, UK), with a 0.2 mm precision and measurement range from 0 to 48 mm, a dermographic pencil, a metal, inextensible and narrow anthropometric tape, model Lufkin W606PM with 0.2 mm precision and a bicondylar pachymeter to measure the diameter of small bones, model Holtain, with 1 mm precision and measuring range from 0 to 48 mm [16].

Blood test and marker analysis: A blood test was carried out at 9 a.m. on an empty stomach, and then the serum was separated from the plasma after centrifuging the samples. PON1 activity was determined using 4-nitrophenyl acetate [17], with an automated clinical chemistry analyser (Olympus A 400, Tokyo, Japan). On the other hand, IL-6 was measured with the ELISA technique (R&D Systems, Madrid, Spain).

Motor Assessment: The 2-min walk test (2MWT) was applied, which determines the functional capacity in people with walking difficulties, especially long distances. In this test, the patients were instructed to “walk at a comfortable pace” back and forth along a 60-foot walkway for 2 min [18]. On the other hand, the 10-m walk test (10MWT) was carried out, which determines the physical function and the general health condition of the patient by means of the walking speed (m/s) [19]. In this test, the participants were told to “walk at a usual comfortable speed” and “as fast as you can without risking falling,” for 10 m. One-meter acceleration and deceleration areas were added to the ends of each 10-m walkway, and the average of three attempts was calculated. In order to assess the functionality of the upper limbs, both on the dominant and non-dominant side, the nine-hole peg test (9-HPT) was used [20]. In this test, the patients sat at a table where there was a shallow container with nine pegs and a plastic board with nine empty holes. All pegs had to be inserted one at a time into the holes and then removed again one at a time and placed into the shallow container. The time to complete the task was recorded twice for both hands and mean values were taken for each hand. Finally, handgrip strength was measured using a handheld Jamar dynamometer (Sammons Preston Inc., Bolingbrook, IL, USA). This is a reliable (ICC values 0.85–0.98) and valid test to measure isometric grip strength in healthy subjects and MS patients [21]. It was performed in an upright sitting position with 90° flexion of the elbow next to the body. Values (kilogram-force) of three maximum voluntary grip strength movements were taken for each hand.

## 3. Presentation of Case Report

### Case Description

The case of a patient was included, alongside her brother, in a randomised double-blind clinical trial for motor disease treatment. The patient and her brother were the only ones from a large sample that had CMT2. In both cases, electrophysiological studies show normal nerve conduction velocities and a decrease in sensitive and motor neurographic responses, in addition to fasciculations, fibrillations and positive waves in electromyography while resting and great motor unit potentials in the one of maximal activation. As far as muscle weakness is concerned, after the exploration prior to the intervention, patients had similar degrees of significant muscular distal amyotrophy to the knee, with a more noticeable weakness especially in plantar and flexor muscles, leading to equinus and varus feet. The genetic cause, in the two study subjects, was mutation in the membrane metalloendopeptidase (MME) gene in homozygosity. As far as the genetic mutation is concerned, the diagnosis report presented by the patients show an alteration in MME gen (p.Arg488Ter, a single nucleotide variant where arginine is changed by a termination codon). However, we do know that they have no familiar history of the disease, being these two siblings the only ones that have developed it among 5 siblings. On the other hand, it is also known the absence of other family members who have developed compatible symptoms with the disease, but there is consanguinity of the parents, which could explain the homozygous mutation with recessive autosomal behaviour.

Regarding the severity levels in both patients, the electrophysiological study and mobility tests showed no significant differences between both individuals. The participant treated with EGCG was a 56-year-old woman, weighing 92.4 kg and 156 cm tall, diagnosed in 2008 with CMT2; she was independent in her daily life, with constant weight gain and without any other medical history of interest. Her brother followed a diet of the same characteristics and a placebo treatment as described previously. He was 51 years old, weighing 111.4 kg and 168 cm tall, diagnosed in 2008. In addition, he was also independent in his daily life, with constant weight gain and without any other medical history of interest.

Before the intervention, the patient who received EGCG had the following anthropometric measurements: waist 104.5 cm, tricipital fold 35 cm, body mass index (BMI) 37 kg/cm^2^, body fat percentage 28.6% and muscle mass percentage 36.78%. These anthropometric aspects exist alongside muscle weakness, spasms and fatigue. The patient who received placebo had the following anthropometric measurements: waist 120.5 cm, tricipital fold 29 mm, BMI 41.6 kg/cm^2^, body fat percentage 27.6% and muscle mass percentage 34.25%.

After the intervention, the values of these parameters were, for the patient who was treated with EGCG: waist perimeter 98 cm (difference of −6.5 cm) tricipital fold 29 mm (difference of −6 mm), BMI 34.45 kg/cm^2^ (difference of −2.55), body fat percentage 24.76% (difference of −3.82%) and muscle mass percentage 39.58% (difference of 2.8%) (Figure 1). The patient who received placebo had: waist perimeter 128 cm (difference of 7.5 cm) tricipital fold 30 mm (difference of 1 mm), BMI 39.3 kg/cm^2^ (difference of −2.3), body fat percentage 29.65% (difference of 1.99%) and muscle mass percentage 31.46% (difference of −2.79%) (Figure 2).

These changes are expressed in weight loss since the initial weight was 92.4 kg and resulted in 86.2 kg (a difference of 6.2 kg), with a decrease in body fat and an increase in muscle mass (Figure 1). However, the patient who received placebo had an initial weight of 118.4 kg and resulted in 112 kg (a difference of 6.4 kg), highlighting the fact that weight loss was due to a decrease in muscle mass, which means there was a loss in strength and muscle power (Figure 2).

These anthropometric characteristics for both patients resulted in improvements in the patient treated with EGCG, yet this did not happen for the patient receiving placebo. In particular, the patient improvement motor skills (in all applied tests) and biochemical markers (where concentration of IL-6 and PON1 in the blood decreased and increased, respectively). Changes in the 2MWT especially stand out, with an increase of 50 m (from 60 to 110 m), or the change in IL-6 in serum which decreased from 4.11 pg/mL to 1.07 pg/mL, respectively, and whose value is estimated within normal values (Table 1).

## 4. Discussion

Once our intervention had been conducted, all the analysed variables worsened in the control patient who received placebo. However, an improvement was observed in the patient who received EGCG, showing an increase in lean mass and a decrease in fat mass, alongside improvements in oxidation and inflammation biomarkers (PON1 and IL-6) and greater motor activity in lower limbs and hands. This could be due to an energy improvement at a mitochondrial level, especially skeletal muscle. In this sense, IL-6 was precisely the first myokine described with a function related to obtaining energy from the muscle [22]. Thus, increased levels are associated with its function of increasing the availability of energy to maintain muscle contraction, by stimulating the metabolism of glucose and lipids [23]. There was also an improvement in subjective perception as the treated patient indicates she feels better, with less muscle aches and can perform daily life activities easier, also feeling less tired when performing said activities.

Our study resulted in observing a large decrease in IL-6 levels by administering high quantities of EGCG for 4 months. Said increase could be on account of an improvement in energy activity in the mitochondria due to the neuroprotective function of EGCG, associated with lower oxidation as indicated in the increase of the PON1 enzyme. Regarding improvements in mitochondrial energy balance, EGCG also improves the activity of the Na(+)/K(+)–ATPase pump that is especially important in muscle and nerve cells in order for correct mitochondrial functioning [24], which is altered in the disease [25]. These results reinforced the promising advances observed in animal models of the disease after administering antioxidants, such as vitamin C and curcumin [26]. Moreover, said results would be in line with improvements obtained in CMT1A after administering another antioxidant and anti-inflammatory with mitochondrial benefits, such as melatonin. Particularly in this last study, improvements in oxidation state and inflammation levels [27] were observed after 3 months of treatment. IL-6 levels were also established within normal values, coinciding with our results after administering EGCG.

## 5. Concluding Remarks

Both at a metabolic and functional level, with a decrease in muscle and increase in inflammation and oxidation affecting motor functioning, administering EGCG shows very good results in all these variables. This evidence makes it a possible therapeutic alternative to treat the disease, which could be interesting to try to improve the functional capacity of these patients. Nevertheless, definitive conclusions cannot be drawn regarding the results of a 4-month intervention, in the case of a progressive chronic disease, therefore, we must continue to delve into the understanding of the impact of this type of treatments in CMT2.

## Figures and Tables

**Figure 1 medicina-57-00104-f001:**
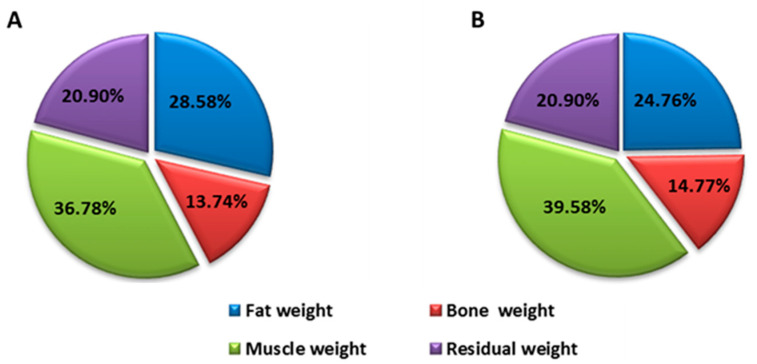
Distribution of body weight before (**A**) and after (**B**) intervention of the patient treated with EGCG.

**Figure 2 medicina-57-00104-f002:**
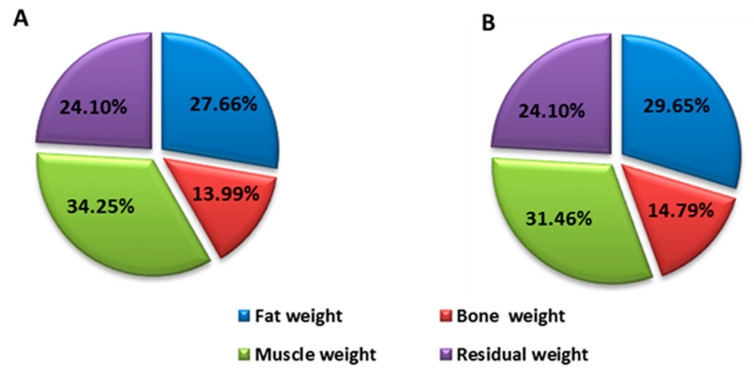
Distribution of the control patient’s body weight before (**A**) and after (**B**) intervention.

**Table 1 medicina-57-00104-t001:** Result of the CMT2 patient treated with EGCG at baseline and follow-up visit.

Variable		CMT2 Patient Treated with EGCG	
		Baseline	Follow-Up
IL-6 (pg/mL)		4.11	1.07
PON1 (UI/L)		3.11	3.23
2MWT (m)		60.00	110.00
Jamar (kg)	Right hand Mean (Dominant)	8.67	9.50
	Left hand Mean	2.67	8.50
9-HPT (s)	Right hand Mean (Dominant)	25.50	24.00
	Left hand Mean	25.00	24.00
10MWT (s)	Self-selected speed Mean	11.33	5.6
	Maximum speed Mean	8.33	6.6

CMT2: Charcot-Marie-Tooth disease type 2; EGCG: epigallocatechin gallate; IL-6: interleukin-6; PON1: paraoxonase 1; 9-HPT: nine-hole peg test; 10MWT: 10-m walk test; 2MWT: 2-min walk test.

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
