# Peer review of "Metabolic and Functional Improvements in a Patient with Charcot-Marie-Tooth Disease Type 2 after EGCG Administration: A Case Report"

_medicina, 2021, doi:10.3390/medicina57020104_

Round 1

Reviewer 1 Report

The authors addressed to the comments and the revised manuscript was far better than the previous one, but still additional major and minor questions are needed to be addressed.

The authors stated a homozygous MME mutation as the genetic cause. CMT patients with the MME mutations usually showed late-onset axonal CMT2 similarly with this study, however they usually showed slightly decreased nerve conduction velocity. In this study, both affected cases atypically showed normal NCVs. May it the identified MME mutation not be a genetic cause? It is recommended to provide followings: (1) What changed nucleotide and amino acid sequences by the mutation? (2) how to find MME mutation? (3) how to confirm the mutation was just causative?, and (4) family history?

For the statement of “Considering the progressive worsening of this disease..”: Although CMT is a degenerative disorder, it is known that it does not progress as rapidly as the severity of symptoms changes after 4 months. It is recommended a simple comparison between before and after treatment in EGCG-treated patients.

Minor points

[L25] inCMT2 > in CMT2

[L32-36] For the last sentence in the Abstract, “Considering the progressive worsening of this disease regarding ... with EGCG are noteworthy and this treatment is proposed as a good candidate to treat the disease.”, it is recommended to separate it into two sentences.

[L44] [1].CMT > [1]. CMT

[L80-83] It is recommended to delete following sentence: In order to determine that the possible improvement ....... the brother of the patient with the same genetic disease was used in this study as a negative control.

[L93] It is recommended to delete following sentence: Furthermore, an ISAK level 3 certified anthropometrist took the measurements [15].

[L95-96] For model SECA 220 Hamburg (Germany) and model Holtain LTD Crymych (UK), provide City names.

[L246] de Boer,J.F.;Perton, F.G.;Annema, .. U.J.Increased > de Boer,J.F.; Perton, F.G.; Annema, .. U.J. Increased

[Abstract and Meathods] 2-Minute Walk Test, 10-Metre Walk Test, Nine-Hole Peg Test >

2-minute walk test, 10-metre walk test, nine-hole peg test

[L128] CMT2 disease > CMT2

[L146, 149] For kg/cm2, “2” should be written by superscript.

[L223-225] For Acknowledgments: Delete followings: In this section you can acknowledge any support given which is not covered by the author contribution or funding sections. This may include administrative and technical support, or donations in kind (e.g., materials used for experiments).

[References] For 6, provide page numbers. For 6, 14, 16, 18,..... , do not use capital letters except for first letters of paper titles.

[Figure 1] It is recommended to delete outer boxes and upper “Weight distribution” in both A and B. Weight fat > Fat weight.

[Figure 2] It is recommended to delete outer boxes and upper Weight distribution in both A and B.

[Table 1] IL-6(pg/ml), PON1(UI/L) > IL-6 (pg/ml), PON1 (UI/L). Spell out CMT2 and EGCG at the bottom of the table.

Reviewer 2 Report

Authors have ansewered and corrected my questions throught the text.

Author Response

Point 1: Authors have ansewered and corrected my questions throught the text.

Response 1: We appreciate it.

Round 2

Reviewer 1 Report

The authors tries to addressfor the reviewer's comments, but several points were not seemed to be properperly addressed.

The author stated that the affected persons have p.Arg488Ter mutation in the MME mutation. I wonder why the authors did not know how find the MME mutation. Anyway, in the result section, provide the p.Arg488Ter homozygous mutation in MME.

For the description of [Abstract] Considering the progressive worsening of this disease... and [Concluding remarks]  Taking into account the progressive worsening of CMT2....., as pointed previously, I don't think that clinical symptoms of the CMT patients are so rapidly worsen just during 4 monthes. It is recommended to delete these expressions.

Figure 1: Weight fat > fat weight

Figure 1 and 2: It is recommended to eleminate puter box in each figure. 

Reference 8: 26,342-348 > 26, 342-348 

Reference 15: 26March > 26 March

Author Response

This manuscript is a resubmission of an earlier submission. The following is a list of the peer review reports and author responses from that submission.

Round 1

Reviewer 1 Report

The study examined effect of epigallocatechin gallate (EGCG) treatment in a CMT2 patient. A woman with CMT2 showed improving effect by 4 months treatment of the EGCG. This study may propose a candidate to treat CMT, however, several points are needed to address.

This study examined in subjects with two siblings with CMT2. But no description was found for the diagnosis of the CMT2. It is recommended to provide electrophysiological values and distal muscle weakness degrees. It is recommended to provide the genetic cause, if available.

This study showed an improvement effect for the EGCG treatment only from an affected woman, although this manuscript is case report. This make the conclusion less convincing.

As the authors stated, CMT is the progressive worsening disease. But the worsening degree may low just during the 4 months duration.

Minor points

No space between two words were found in many places. Just at abstract: (CMT2)treated, (EGCG)for, inCMT2, mgof, paticipantswere.

Insert a space between numerical and unit, such as 150-200kg > 150-200 kg, 0.2mm > 0. Mm, 156cm >156 cm…

Provide city and country names for the companies, such as (Olympus A 400), and (R&D Systems).

Line 22: CMT2 disease > CMT2

Line 63: Provide full words for ALS.

Reviewer 2 Report

 This is a prospective study on a patient with HMSN type II who was given epigallocatechin gallate EGCG for 4 months compared to another patient who was given a placebo.

There are several drawbacks in this manuscript. The co-existing of CIDP or any inflammatory polyneuropathy with HMSN is doubtful, according to current literature. Few case-repots on this topic suggested that at a certain time point an inflammatory reaction affecting the nerves is possible in hereditary neuropathies but this does not justify the existence of a continuum course or relapsing-remitting course of auto-immune neuropathy.

One cannot draw conclusions on the results of 4-months treatment in case of chronic progressive disease. Variability of measurements during this short time period should be attributed to chance and not to disease progression or improvement.

Both subjects are obese, in addition to Charcot-Marie-Tooth neuropathy. It is not clear whether EGCG targeting weight loss or neuropathy improvement.

There is no evidence .i.e. neurological signs or neurophysiological abnormalities that the two subjects suffered from HMSN of similar severity. One could assume that the bother, who was given the placebo, had a more severe form of neuropathy from the onset of the observation period.

The conclusion “improve the prognosis” is too strong and based on secondary -supplementary measurements

Reviewer 3 Report

Concerns,

English should be carefully checked.

The acceptance of the paper for this reviewer after minor grammar corrections is clearly conditioned with the felling that authors are performing the same therapeutic experiment described here but in a statistically confident population